# TRAINING-FREE GENERATIVE MODELING BY KERNELIZING PRETRAINED DIFFUSION MODELS

## ABSTRACT

Generative diffusion models, including stochastic interpolants and score-based approaches, require learning time-dependent drift or score functions through expensive neural network training. Here we avoid these computations by representing the drift in a reproducing kernel Hilbert space, reducing the learning problem to solving linear systems. The key challenge becomes selecting kernels with sufficient expressiveness for the drift learning task. We address this by constructing kernels from pretrained drift or score functions, leveraging the fact that our linear systems depend only on gradients of kernel features—not the features themselves. Since pretrained drifts provide these gradients directly, we can build expressive kernels without access to the underlying feature representations. This enables seamless combination of multiple pretrained models at inference time and cross-domain enhancement through the same framework. Experiments demonstrate competitive sample quality with significantly reduced computation, consistent ensemble improvements, and successful cross-domain enhancement—even cheap, low-quality models can match expensive high-quality models when combined through our framework.

## 1 INTRODUCTION

The rapid advancement of generative diffusion models has led to an explosion of pretrained models across diverse domains and modalities, ranging from high-quality models requiring substantial computational resources to more accessible models trained with limited budgets. This proliferation presents both new challenges and unprecedented opportunities: how can we effectively leverage and combine these existing models without expensive retraining? While current approaches to model combination often require specialized training procedures, domain-specific architectures, or handcrafted weight interpolation schemes that may degrade individual model performance, the growing ecosystem of pretrained models—including computationally accessible "weak" models—opens possibilities for novel combination strategies that can achieve high-quality generation from low-cost components.

However, realizing these opportunities requires careful rethinking how models are combined. Traditional ensemble methods either average parameters (potentially destroying specialized knowledge) or require running multiple models simultaneously (multiplying computational costs). Recent approaches like mixture of experts require architectural redesign, while parameter-efficient methods face degradation issues when combining multiple adaptations. Rather than operating in weight space (where models may be incompatible) or requiring simultaneous inference (which is computationally expensive), this suggests combining models in function space through kernel representations. This preserves individual model strengths while enabling training-free combination.

Following this reasoning, this paper introduces a new approach: **kernelizing pretrained diffusion models for training-free generative modeling**. Our first key insight is that the stochastic interpolant framework can be reformulated as a kernel learning problem, where expensive neural network training is replaced by solving linear systems to estimate the drift needed in the diffusion used

as generative model. Under characteristic kernel assumptions, we provide conditions where this method recovers the exact drift, using maximum mean discrepancy as a goodness-of-fit indicator.

In this approach, the challenge reduces to identifying expressive kernels for representing this drift. Our second key insight is that pretrained diffusion models can be directly used to construct these kernels, bypassing the need for access to underlying feature representations. This rests on a crucial observation: the linear systems depend only on gradients of kernel features, not the features themselves. Since pretrained diffusion models already provide these gradients (as scores or drifts), we can build expressive kernels without knowing the original feature space. This enables seamless model combination through linear algebra rather than neural network training.

While our framework applies broadly to model combination problems, we demonstrate its effectiveness through two key application domains. First, **mixture of weak experts**: combining multiple models trained on the same dataset but with limited computational resources—for instance due to early stopping, reduced capacity, or constrained training time. While each weak model individually underperforms, they collectively contain complementary information that, when properly combined, can recover the performance of a fully-trained strong model. Second, **cross-domain enhancement**: combining models trained on different datasets to improve generation quality for each individual domain. For instance, a model trained on natural images might enhance the generation quality of a face-specific model when targeting face generation, and vice versa.

Our main contributions can be summarized as follows:

- We reformulate stochastic interpolant learning as a kernel method where the drift is obtained via direct solutions of linear systems (Theorem 2.5).

- We establish conditions under which this kernel approach recovers the true drift functions under characteristic kernel assumptions.

- We show how pretrained scores/drifts can construct kernels that preserve the expressiveness of the original models, enabling two key capabilities:

- *Training-free model combination* that preserves individual model architectures while enabling single-model inference, unlike mixture-of-experts architectures that require specialized training or ensemble methods that multiply inference costs.

- *Cross-domain enhancement* where models trained on different datasets can be combined to mutually improve generation quality for each individual domain, rather than traditional domain adaptation which transfers from source to target.

- We demonstrate the applicability of the approach through experiments on MNIST and CelebA where we show that competitive sample quality can be obtained with significantly reduced computational requirements, and observe consistent improvements from model ensembling and successful cross-domain enhancement.

The method addresses several key challenges in modern generative modeling: the computational expense of training from scratch, the difficulty of combining specialized models, and the challenge of adapting to new domains with limited data. It also offers a path toward **computational democratization**: Training many weak models is more accessible than training single strong models, yet our approach shows that several weak models can recover strong model performance.

## 1.1 RELATED WORK

Our approach builds upon recent advances in generative modeling based on dynamical transport of measure, as implemented in score-based diffusion models (Song et al., 2020), flow matching (Lipman et al., 2022), rectified flows (Liu et al., 2022b), and stochastic interpolants (Albergo & Vanden-Eijnden, 2022; Albergo et al., 2023). Existing work in this area mostly focuses on training single models rather than combining existing ones. We reformulate the stochastic interpolant framework as a **kernel method** (Muandet et al., 2017), enabling training-free estimation of the drift through

linear solve rather than expensive neural network training and provding theoretical guarantees using the mean maximum distance (MMD). RKHS reformulations of diffusion models have been considered e.g. in Maurais & Marzouk (2024); Yi et al. (2024): in contrast with these approaches that use standard kernels (Gaussian/RBF, Laplacian, etc.) we build the kernel using pre-trained score or drifts, thereby avoiding the identification of the feature map as well as the calculation of their gradient. **MMD-based methods** in generative modeling have been primarily used to assess performance (Gretton et al., 2012; Sutherland et al., 2017), though recently they also have been leveraged for model-building, see e.g. Galashov et al. (2024); Bortoli et al. (2025); Shen et al. (2025); Zhou et al. (2025). However these methods also require careful engineering of feature maps as opposed to combination of existing scores or drifts.

**Efficient finetuning** methods like LoRA (Hu et al., 2021; Ruiz et al., 2023) leverage the vast amounts of information encoded into the model during pre-training, but typically build on top of a single model and work best in specialising for a specific data property. **Ensembling** models trained on similar data is a well-established way of improving performance. More diverse models yield even better results (Fort et al., 2019) aligning with our mixture of weak experts scenario where models trained with limited resources naturally exhibit diversity yet collectively recover strong performance. However, aggregating model outputs comes at an additional inference cost. In turn, inference-efficient methods that average in weight space, such as Model Soup (Wortsman et al., 2022) and Diffusion Soup (Biggs et al., 2024), can lead to performance degradation. Knowledge transfer from multiple models into one (Hinton et al., 2015) and teacher-student setups (Medvedev et al., 2025) still require training, while parameter-based knowledge transfer requires a careful probing of the model internals (Zhong et al., 2023). In contrast, our method is training-free and achieves cheap inference while maintaining generation quality.

Composing separate models can also be used to enable generalization to different, more complex data than seen in training (Luo et al., 2025; Liu et al., 2021) or for increased generation control (Ho & Salimans, 2022). Frameworks focusing on **enabling compositionality** through distributions did so through the lens of energy based models (Liu et al., 2021; Du et al., 2020) or diffusion models (Liu et al., 2022a). Ensuring that the reverse process samples correctly from the composed model may require complex sampling modifications, such as MCMC (Du et al., 2023) or advanced resampling methods (Skreta et al., 2025; 2024) or precise specifications on how to modify the base distributions (Bradley et al., 2025). Other work enables compositionality through the LoRAs. But linearly merging different LoRAs (Shah et al., 2024; Zhang et al., 2023; Huang et al., 2023) shows reduced accuracy with an increasing number of models. A more recent approach (Zhong et al., 2024) presents a more accurate training-free approach, but lacks a firm theoretical underpinning. Our work presents a theoretically grounded method with a simple inference procedure.

## 2 Generative Models with Kernelized Stochastic Interpolants

### 2.1 Feature Maps, Kernels, and Maximum Mean Discrepancy

We begin by giving some background material on kernel methods with feature maps, referring the reader to Appendix A for more details.

**Definition 2.1** (Feature Map and Kernel). *A **feature map** is a function $\Phi : \mathcal{H} \to \mathcal{F}$ that maps points from the input space $\mathcal{H}$ to the feature space $\mathcal{F}$, where $\mathcal{F}$ is a Hilbert space with inner product $\langle \cdot, \cdot \rangle_{\mathcal{F}}$. Associated with $\Phi : \mathcal{H} \to \mathcal{F}$, there is a **positive-definite kernel function** $k : \mathcal{H} \times \mathcal{H} \to \mathbb{R}$ given by*

$$k(x, y) = \langle \Phi(x), \Phi(y) \rangle_{\mathcal{F}} \tag{1}$$

Correspondingly, given any positive-definite kernel $k$, the representation (1) for some feature map $\Phi$ is guaranteed for all positive-definite kernels by Mercer's theorem.

**Definition 2.2** (Maximum Mean Distance (MMD)). *Given a positive-definite kernel $k(x, y)$ defined by the feature map $\Phi$, the maximum mean distance between two distribution $\mu$ and $\hat{\mu}$ with support $\mathcal{H}$ is defined via*

$$\text{MMD}^2(\mu, \hat{\mu}) = \int_{\mathcal{H} \times \mathcal{H}} (\mu(dx) - \hat{\mu}(dx))k(x, y)(\mu(dy) - \hat{\mu}(dy))$$

$$= \left\| \mathbb{E}_{x \sim \mu}[\Phi(x)] - \mathbb{E}_{y \sim \hat{\mu}}[\Phi(y)] \right\|_{\mathcal{F}}^2 \tag{2}$$

*where $\| \cdot \|$ denotes the norm on $\mathcal{F}$.*

We also have

**Definition 2.3** (Characteristic Kernel). *The kernel $k : \mathcal{H} \times \mathcal{H} \to \mathbb{R}$ is **characteristic** iff the MMD is a true distance: $\text{MMD}(\mu, \hat{\mu}) = 0$ implies that $\mu = \hat{\mu}$.*

Examples of characteristic kernels are given in Appendix A where we also give a more detailed definition of these kernels.

## 2.2 KERNELIZED STOCHASTIC INTERPOLANTS

Our aim is to construct a generative model to sample a target distribution $\mu$ known through data. We will do so using the framework of stochastic interpolants:

**Definition 2.4.** *Given the data $a \in \mathcal{H}$ with probability distribution $\mu$ and some independent Gaussian variable $z \in \mathcal{H}$ with probability distribution $\gamma = \mathsf{N}(0, Id)$, the **stochastic interpolant** between $z$ and $a$ is the stochastic process*

$$I_t = \alpha_t z + \beta_t a, \qquad z \sim \gamma, \quad a \sim \mu, \quad a \perp z, \quad t \in [0, 1] \tag{3}$$

*where $\alpha, \beta \in C^1([0, 1])$ with $\dot{\alpha}_t < 0$, $\dot{\beta}_t > 0$, $\alpha_0 = \beta_1 = 1$, and $\alpha_1 = \beta_0 = 0$.*

For example we could take $\alpha_t = 1 - t$, $\beta_t = t$. By definition, we see that $I_0 = z \sim \gamma$ and $I_1 = a \sim \mu$, i.e. this process interpolates between Gaussian samples from $\gamma$ at time $t = 0$ and target samples from $\mu$ at that time $t = 1$.

From the results in Albergo & Vanden-Eijnden (2022); Albergo et al. (2023), we know that, for each $t \in [0, 1]$, the law of $I_t$ is the same as the law of the solution $X_t$ of the stochastic differential equation

$$dX_t = b_t(X_t)dt + \epsilon_t s_t(X_t)dt + \sqrt{2\epsilon_t}dW_t, \qquad X_{t=0} = z \sim \nu, \tag{4}$$

Here $\epsilon_t \geq 0$ is a diffusion coefficient that can be adjusted, and $b_t : \mathcal{H} \to \mathcal{H}$ and $s_t : \mathcal{H} \to \mathcal{H}$ are the velocity field and the score given by

$$b_t(x) = \mathbb{E}[\dot{I}_t | I_t = x] = \dot{\alpha}_t \mathbb{E}[z | I_t = x] + \dot{\beta}_t \mathbb{E}[a | I_t = x], \qquad s_t(x) = -\alpha_t^{-1} \mathbb{E}[z | I_t = x], \tag{5}$$

where $\dot{I}_t = \dot{\alpha}_t z + \dot{\beta}_t a$ is the derivative of $I_t$ with respect to $t$ (e.g. $\dot{I}_t = a - z$ if $\alpha_t = 1 - t$, $\beta_t = t$) and $\mathbb{E}[\cdot | I_t = x]$ denotes the expectation over the law of $I_t$ conditional on $I_t = x$. Using the relation $x = \mathbb{E}[I_t | I_t = x] = \alpha_t \mathbb{E}[z | I_t = x] + \beta_t \mathbb{E}[a | I_t = x]$ the score can be expressed in term of the velocity and vice versa via

$$s_t(x) = \frac{\beta_t b_t(x) - \dot{\beta}_t x}{\alpha_t(\alpha_t \dot{\beta}_t - \dot{\alpha}_t \beta_t)} \quad \Leftrightarrow \quad b_t(x) = \frac{\alpha_t s_t(x) + \dot{\beta}_t x}{\beta_t(\alpha_t \dot{\beta}_t - \dot{\alpha}_t \beta_t)} \tag{6}$$

showing that we only need to estimate $b_t(x)$ or $s_t(x)$.

In general, neither $b_t(x)$ nor $s_t(x)$ are available in closed form. However, our next reult shows that *we can give exact tractable expressions for them using any feature map that leads to a characteristic kernel*:

**Theorem 2.5.** *Assume that the kernel $k$ is characteristic in the sense of Definition 2.3 and define the operator $K_t \in \mathcal{F} \times \mathcal{F}$ via*

$$K_t = \mathbb{E}\big[\langle \nabla\Phi(I_t), \nabla\Phi(I_t)\rangle\big] \tag{7}$$

*where $\nabla\Phi(x)$ denotes the gradient of $\Phi(x)$ with respect to $x$, $\langle\cdot,\cdot\rangle$ is the inner product in the input space $\mathcal{H}$, and $\mathbb{E}$ denotes expectation over the law of $I_t$. Assume that $K_t$ is positive-definite for all $t \in [0,1]$. Then, the velocity field $b_t(x) = \mathbb{E}[\dot{I}_t | I_t = x]$ can be expressed for each $t \in [0,1]$ as*

$$b_t(x) = \big\langle \nabla\Phi(x), \eta_t\big\rangle_{\mathcal{F}}, \tag{8}$$

*where $\eta_t \in \mathcal{F}$ is the unique solution to the linear system of equations*

$$\langle K_t, \eta_t\rangle_{\mathcal{F}} = \mathbb{E}\big[\langle \nabla\Phi(I_t), \dot{I}_t\rangle\big] \tag{9}$$

*Similarly, the score $s_t(x) = -\alpha_t^{-1}\mathbb{E}[z | I_t = x]$ can be expressed for each $t \in (0,1]$ as*

$$s_t(x) = -\alpha_t^{-1}\big\langle \nabla\Phi(x), \zeta_t\big\rangle_{\mathcal{F}}, \tag{10}$$

*where $\zeta_t \in \mathcal{F}$ is the unique solution to the linear system of equations*

$$\langle K_t, \zeta_t\rangle_{\mathcal{F}} = \mathbb{E}\big[\langle \nabla\Phi(I_t), z\rangle\big] \tag{11}$$

Note that all the expectations in the theorem can be estimated empirically on the data, making the solution of the linear systems (9) and (11) tractable, and expressions (8) and (10) for $b_t(x)$ and $s_t(x)$ usable in the SDE (4). Note also that we can still get the score $s_t(x)$ from $b_t(x)$ and vice versa via (6), so only one of them needs to be estimated.

The proof of Theorem 2.5 is given in Appendix B. To give some intuition for (8), recall that, by the $L^2$ characterization of the conditional expectation, the velocity $b_t$ defined in (5) is the mimimizer over $\hat{b}_t$ of the objective function

$$L_b[\hat{b}_t] = \mathbb{E}\big[\big\|\hat{b}_t(I_t) - \dot{I}_t\big\|^2\big] \tag{12}$$

where the expectation is taken over the law of $I_t$. If we insert the ansatz $\hat{b}_t(x) = \langle \nabla\Phi(x), \hat{\eta}_t\rangle_{\mathcal{F}}$ into (12) we turn this objective into a quadratic objective for $\hat{\eta}_t \in \mathcal{F}$:

$$\begin{aligned} L_b[\langle \nabla\Phi, \hat{\eta}_t\rangle_{\mathcal{F}}] &= \mathbb{E}\big[\big\|\langle \nabla\Phi(I_t), \hat{\eta}_t\rangle_{\mathcal{F}} - \dot{I}_t\big\|^2\big] \\ &= \big\langle \hat{\eta}_t, K_t\hat{\eta}_t\big\rangle_{\mathcal{F}} - 2\big\langle \mathbb{E}\big[\langle \nabla\Phi(I_t), \dot{I}_t\rangle\big], \hat{\eta}_t\big\rangle_{\mathcal{F}} + \mathbb{E}[\|\dot{I}_t\|^2]. \end{aligned} \tag{13}$$

where $K_t$ is the operator defined in (7). Assuming that $K_t$ is postive-definite, the minimizer of this objective is unique and given by the unique solution $\eta_t$ of (9). This establishes that (8) is the best approximation of $b_t(x)$ in the class $\hat{b}_t(x) = \langle \nabla\Phi(x), \hat{\eta}_t\rangle_{\mathcal{F}}$. Theorem 2.5 shows that, if the kernel is characteristic, this class is expressive enough to recover the exact $b_t(x)$. A similar argument can be used to justify (10) starting from the denoising loss for $s_t(x)$,

$$L_s[\hat{s}_t] = \mathbb{E}\big[\big\|\hat{s}_t(I_t) + \alpha_t^{-1}z\big\|^2\big] \tag{14}$$

and minimizing it over function in the class $\hat{s}_t(x) = \langle \nabla\Phi(x), \hat{\zeta}_t\rangle_{\mathcal{F}}$.

*We stress that* (8) *and* (10) *remain usable approximations of $b_t(x)$ and $s_t(x)$ regardless of whether the kernel is characteristic.* Since we do not explicitly construct the kernel in our approach, we cannot guarantee this property (see however the remark below after (16)), but (8) and (10) remain practically viable as demonstrated through our numerical experiments.

---

**Algorithm 1:** Learning-Free Generation with Pre-Trained Velocity Fields

**input:** time step $h = 1/K$ with $K \in \mathbb{N}$, batch size $N \in \mathbb{N}$, pretrained velocity fields $b_t^i(x)$,
      $i = 1, \ldots, P$, coefficients $\alpha_t, \beta_t, \epsilon_t \geq 0$

**initialize:** $X_0 \sim \gamma$;

**for** $\underline{k = 0, \ldots, K - 1}$ **do**

    set $t = kh$;

    draw samples $\{z_1, \ldots, z_N\} \sim \nu$, and $\{a_1, \ldots, a_N\} \sim \mu$;

    calculate $I_t^n = \alpha_t z_n + \beta_t a_n$ and $\dot{I}_t^n = \dot{\alpha}_t z_n + \dot{\beta}_t a_n$, $n = 1, \ldots, N$;

    obtain $(\eta_t^1, \ldots, \eta_t^P)$ by solving the linear system:

$$\sum_{j=1}^{P} \frac{1}{N} \sum_{n=1}^{N} \langle b_t^i(I_t^n), b_t^j(I_t^n) \rangle \eta_t^j = \frac{1}{N} \sum_{n=1}^{N} \langle b_t^i(I_t^n), \dot{I}_t^n \rangle, \quad i = 1, \ldots, N$$

    set $\qquad b_t(X_t) = \sum_{i=1}^{P} b_t^i(X_t)\eta_t^i \qquad$ and $\qquad s_t(X_t) = \dfrac{\beta_t b_t(X_t) - \dot{\beta}_t X_t}{\alpha_t(\alpha_t \dot{\beta}_t - \dot{\alpha}_t \beta_t)}$;

    update $\qquad X_{t+h} = X_t + h\left(b_t(X_t) + \epsilon_t s_t(X_t)\right) + \sqrt{2h\epsilon_t}\, g_t, \qquad g_t \sim \mathsf{N}(0, \mathrm{Id})$

**output:** $X_1 \sim \mu$ (approximately)

---

## 2.3 PRACTICAL CONSIDERATIONS

Theorem 2.5 shows that we can use any characteristic kernel to construct a training-free generative model. Still, in practice, the choice of the kernel will dramatically impact the performance of this model. Here we discuss a simple and natural way to *use kernels without having to identify their feature map explicitly using pre-trained models*. To this end, a key observation is that:

> Neither equations (8) and (9) defining $b_t(x)$, nor equations (10) and (11) defining $s_t(x)$ depend on the feature map $\Phi(x)$ itself: rather they only depend on its gradient $\nabla\Phi(x)$.

This indicates that we can pick $\nabla\Phi(x)$ directly rather than $\Phi(x)$, thereby avoiding the costly computation of a gradient. One way to do so is to use pretrained velocity fields. Denoting these velocity by $b_t^i(x)$ with $i = 1, \ldots, P$, this amounts to assuming that the feature space $\mathcal{F}$ is $N$-dimensional, and that the $i$th component of $\nabla\Phi(x)$ is[1]

$$\nabla\Phi_i(x) = b_t^i(x), \qquad i = 1, \ldots, P \tag{15}$$

This makes the feature map time-dependent, but does not change the result of Theorem 2.5 since (8) and (9) as well as (8) and (9) holds pointwise in time. For example, if we use (15) in (8) and (9), these equations become

$$b_t(x) = \sum_{i=1}^{P} b_t^i(x)\eta_t^i \quad \text{where} \quad \sum_{j=1}^{P} \mathbb{E}\left[\langle b_t^i(I_t), b_t^j(I_t) \rangle\right]\eta_t^i = \mathbb{E}\left[\langle b_t^i(I_t), \dot{I}_t \rangle\right]. \tag{16}$$

---

[1]Strictly speaking, the velocity fields $b_t^i(x)$ and the scores $s_t^i(x)$ should be in gradient form to use representations (15) and (17). For scores, this holds by definition since $s_t(x) = \nabla \log \rho_t(x)$. For velocity fields, any $b_t(x)$ can in principle be expressed as $\nabla\phi_t$ for some potential $\phi_t$ (by solving the appropriate Poisson equation to preserve the transport dynamics). While pretrained models are not typically trained to enforce this gradient structure explicitly, in practice we observe that using them directly in our framework works effectively, indicating that our approach remains robust even when the theoretical gradient representation is not rigorously maintained.

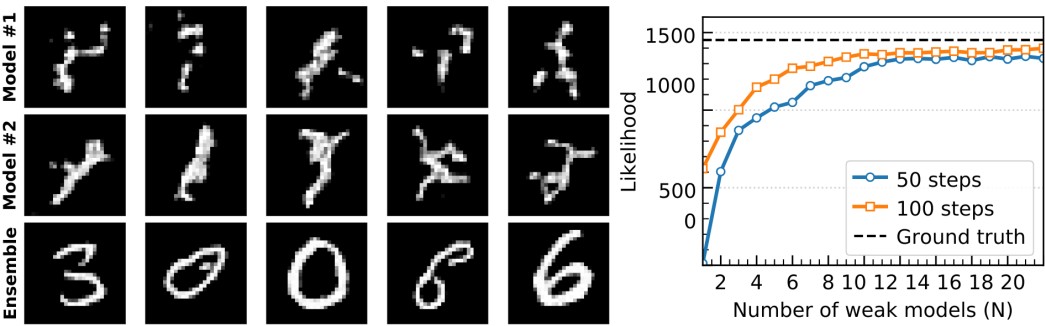

Figure 1: **Left:** Sample images from models trained on MNIST. First and second rows: samples from weak models trained with 50 and 100 SGD steps, respectively. Third row: samples from our approach kernelizing 20 weak models trained over 100 steps. **Right:** Ground truth likelihood averaged over 10k samples from our approach as a function of the number of weak models used.

Having calculated $b_t(x)$ this way, we can then estimate $s_t(x)$ using (6). In terms of generation, this leads to the scheme summarized in Algorithm 1. *Note that the linear solves required in this algorithm need to be performed only once, and can be re-used for each sample generation—the cost of this operation depends on the number $P$ of pretrained models used, which can be kept relatively small as demonstrated in our numerical experiments.* Note also that we could guarantee that the kernel is characteristic by using e.g. the feature map of a characteristic kernel (Gaussian/RBF, Laplacian) and adding its gradient to (15): in our experiments, we observed that such an addition was unnecessary.

Similarly if $s_t^i(x)$ with $i = 1, \ldots, P$ are pretrained scores, one can use

$$\nabla \Phi_i(x) = s_t^i(x), \qquad i = 1, \ldots, P \tag{17}$$

In this case (10) and (11) become

$$s_t(x) = -\alpha_t^{-1} \sum_{i=1}^{P} s_t^i(x) \zeta_t^i \quad \text{where} \quad \sum_{j=1}^{P} \mathbb{E}\big[\langle s_t^i(I_t), s_t^j(I_t)\rangle\big] \zeta_t^i = \mathbb{E}\big[\langle s_t^i(I_t), z\rangle\big]. \tag{18}$$

and we can estimate $s_t(x)$ from $b_t(x)$ using (6).

We could also use (15) in the equations for the score $s_t(x)$, or (17) in the equations for the velocity field $b_t(x)$, or combine both feature maps, etc. In the next section we illustrate how to instantiate this construction on concrete examples.

## 3 NUMERICAL ILLUSTRATIONS

### 3.1 MIXTURE OF (WEAK) EXPERTS

We evaluate our method on two datasets: MNIST, with images of size $28 \times 28$ and CelebA, resized to $128 \times 128$. Details of the experimental setup can be found in Appendix C.

**MNIST:** We stress-test our framework in a compute-constrained regime by composing the ensemble from deliberately under-trained models. To this end we train two cohorts: 20 models trained for 50 steps of stochastic gradient descent (SGD) and another 20 trained for 100 steps, each using mini-batches of 128. To induce diversity, each model is initialized with an independent random seed, ensuring that after $n$ SGD updates the members remain distinct. All models share the same

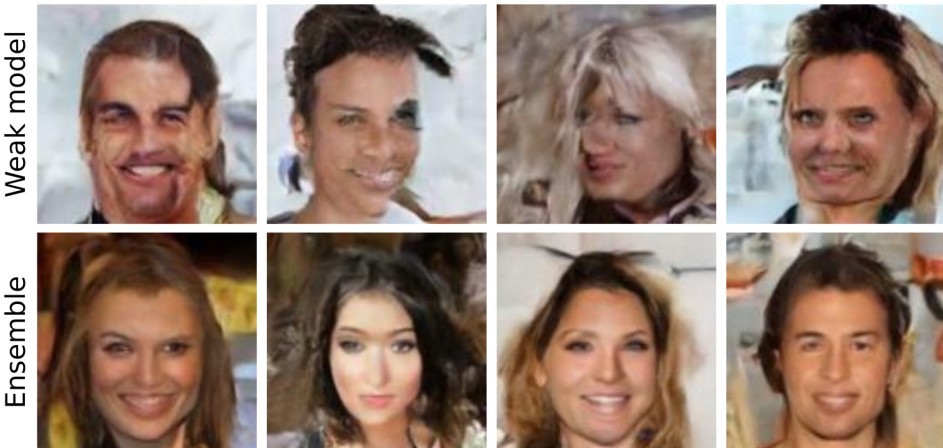

Figure 2: **Top:** Typical samples images from weak model. **Bottom:** Typical samples images from ensemble model of 25 weak models.

backbone: a standard time-conditioned convolutional U-Net described in Appendix C. We then generate samples from the ensemble using Algorithm 1 on the test set.

Figure 1 shows the results from two individual weak models and from an ensemble of 20 weak models (each trained for 100 steps of SGD) used in Algorithm 1. The images on the left panel indicate that our method achieves good visual quality by combining even very weak models. To quantify the performance, we compute the sample likelihoods using a fully trained U-Net as an oracle model, treating its outputs on the test set as ground truth. The plot on right panel shows that the likelihood improves as more weak models are ensembled, but saturates beyond a certain ensemble size: adding additional models yields diminishing returns once this plateau is reached. Using stronger weak models further improves performance and reduces the number of models needed.

**CelebA:** To assess scalability to natural images, we repeat the protocol on CelebA. We train 25 deliberately under-trained models for 5 epochs (learning rate $10^{-4}$, batch size 128) on the training split, each initialized with an independent random seed. All models share the same U-Net backbone described in Appendix C. At evaluation, we generate samples with Algorithm 1. Figure 2 qualitatively demonstrate the improvement performance our approach provides using the 25 weak models.

### 3.2 CROSS-DOMAIN ENHANCEMENT

We test whether ensembles of weak learners trained on semantically related source domains can bootstrap sampling on a distinct target domain. Concretely, we train 10 weak models each on Fashion-MNIST, EMNIST (letters only), MNIST and Kuzushiji-MNIST using 50 steps of SGD with mini-batches of 128 and independent random initializations. All models share the U-Net backbone described in Appendix C. At inference, we use these 30 source-domain models in Algorithm 1 to sample from the MNIST distribution. Figure 3 presents representative samples, illustrating that semantically aligned sources can be composed to produce better target-domain images.

## 4 CONCLUSION

This paper introduces a novel framework for training-free generative modeling by reformulating stochastic interpolants as kernel methods and leveraging pretrained diffusion models to construct expressive kernels. Our approach enables seamless combination of multiple pretrained models without

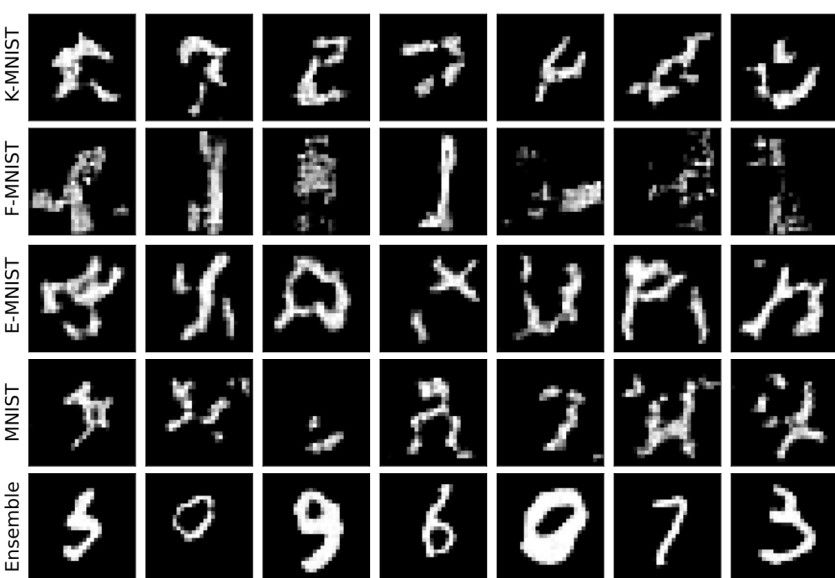

Figure 3: Samples images of weak models trained on Kuzushiji-MNIST (first row), Fashion-MNIST (second row), EMNIST letters (third row) and MNIST (fourth row). The last row shows samples images from MNIST using the weak models from all the source-domains in Algorithm 1.

requiring retraining or architectural modifications, addressing key challenges in the growing ecosystem of specialized generative models.

The theoretical foundation rests on Theorem 2.5, which shows that drift and score functions can be exactly recovered through linear systems when using characteristic kernels. By constructing kernels from pretrained velocity fields or scores, we bypass the need for explicit feature map construction while preserving the expressiveness of the original models. This enables two key capabilities: mixture of weak experts that can recover strong model performance from computationally accessible components, and cross-domain enhancement where models trained on different datasets mutually improve generation quality.

Our experimental results on MNIST and CelebA demonstrate the practical viability of the approach. We observe consistent improvements from ensembling weak models and successful cross-domain enhancement, achieving competitive sample quality with significantly reduced computational requirements compared to training strong models from scratch..

Future work could investigate weaker sufficient conditions for theoretical guarantees, develop more sophisticated constructions using pretrained velocity fields and scores, and extend the framework to other generative modeling tasks.

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

# A  KERNEL METHODS, RKHS, AND MAXIMUM MEAN DISCREPANCY (MMD)

Here we give some additional information about kernel methods. This is a well-mapped topic, with many good references including Muandet et al. (2017) where the interested reader will find more details.

## A.1  KERNEL DEFINITION VIA FEATURE MAPS

A positive-definite kernel function $k : \mathcal{H} \times \mathcal{H} \to \mathbb{R}$ can be defined via a feature map $\Phi : \mathcal{H} \to \mathcal{F}$, where $\mathcal{F}$ is a Hilbert space with inner product $\langle \cdot, \cdot \rangle_{\mathcal{F}}$:

$$k(x, y) = \langle \Phi(x), \Phi(y) \rangle_{\mathcal{F}} \tag{19}$$

This representation is guaranteed for all positive-definite kernels by Mercer's theorem. The function $\Phi$ maps points from the input space $\mathcal{H}$ to the feature space $\mathcal{F}$ where the inner product between mapped points corresponds to the kernel evaluation.

## A.2  REPRODUCING KERNEL HILBERT SPACE (RKHS)

Given a positive definite kernel $k$, there exists a unique Reproducing Kernel Hilbert Space (RKHS) $\mathcal{H}_k$ consisting of the functions $f : \mathcal{H} \to \mathbb{R}$ with the following key property:

$$f(x) = \langle f, k(x, \cdot) \rangle_{\mathcal{H}_k}, \quad \forall f \in \mathcal{H}_k, \ \forall x \in \mathcal{H}, \tag{20}$$

where $k(x, \cdot)$ represents the function that maps $y \in \mathcal{H} \mapsto k(x, y) \in \mathbb{R}$ for each fixed $x \in \mathcal{H}$. This is known as the **reproducing property**. The RKHS $\mathcal{H}_k$ can be constructed as the completion of the space of all finite linear combinations of the form:

$$f(x) = \sum_{i=1}^{n} \alpha_i k(x_i, x), \quad \alpha_i \in \mathbb{R}, \ x_i \in \mathcal{H} \tag{21}$$

The inner product in $\mathcal{H}_k$ is defined such that:

$$\langle k(x, \cdot), k(y, \cdot) \rangle_{\mathcal{H}_k} = k(x, y) \tag{22}$$

## A.3  MEAN EMBEDDING OF DISTRIBUTIONS

For a probability distribution $\mu$ on $\mathcal{H}$, we can define its mean embedding $m_\mu$ in the RKHS $\mathcal{H}_k$ as:

$$m_\mu = \int_{\mathcal{H}} k(x, \cdot) \mu(dx) \tag{23}$$

If we define the feature map $\Phi : \mathcal{H} \to \mathcal{H}_k$ as $\Phi(x) = k(x, \cdot)$, this embedding can also be written as

$$m_\mu = \int_{\mathcal{H}} \Phi(x) \mu(dx) \tag{24}$$

The mean embedding has the property that for any function $f \in \mathcal{H}_k$:

$$\mathbb{E}_{x \sim \mu}[f(x)] = \langle f, m_\mu \rangle_{\mathcal{H}_k} \tag{25}$$

Given samples $\{a_1, a_2, \ldots, a_N\}$ from distribution $\mu$, the empirical estimate of the mean embedding is:

$$\hat{m}_\mu = \frac{1}{N} \sum_{n=1}^{N} k(a_n, \cdot) = \frac{1}{N} \sum_{n=1}^{N} \Phi(a_n) \tag{26}$$

### A.4 MAXIMUM MEAN DISCREPANCY (MMD)

The Maximum Mean Discrepancy (MMD) between two distributions $\mu$ and $\hat{\mu}$ is defined as the RKHS distance between their mean embeddings:

$$\text{MMD}(\mu, \hat{\mu}) = \|m_\mu - m_{\hat{\mu}}\|_{\mathcal{H}_k} \tag{27}$$

Using the properties of inner products, this can be expanded as:

$$\text{MMD}^2(\mu, \hat{\mu}) = \|m_\mu - m_{\hat{\mu}}\|_{\mathcal{H}_k}^2 \tag{28}$$

$$= \langle m_\mu, m_\mu \rangle_{\mathcal{H}_k} + \langle m_{\hat{\mu}}, m_{\hat{\mu}} \rangle_{\mathcal{H}_k} - 2\langle m_\mu, m_{\hat{\mu}} \rangle_{\mathcal{H}_k} \tag{29}$$

$$= \mathbb{E}_{x,x'\sim\mu}[k(x,x')] + \mathbb{E}_{y,y'\sim\hat{\mu}}[k(y,y')] - 2\mathbb{E}_{x\sim\mu, y\sim\hat{\mu}}[k(x,y)] \tag{30}$$

The MMD is a pseudometric on the space of probability distributions. For it to be a true metric (so that $\text{MMD}(\mu, \hat{\mu}) = 0$ if and only if $\mu = \hat{\mu}$), the kernel must be **characteristic**:

**Definition A.1** (Characteristic kernels). *A kernel $k$ is **characteristic** if the mean embedding map $\mu \mapsto m_\mu$ is injective, meaning that different distributions get mapped to different elements in the RKHS. Formally, a kernel is characteristic if:*

$$m_\mu = m_{\hat{\mu}} \quad \Rightarrow \quad \mu = \hat{\mu} \tag{31}$$

For shift-invariant kernels $k(x,y) = k(x-y)$, a sufficient condition for the kernel to be characteristic is that the support of its Fourier transform spans the entire domain. Examples of shift-invariant characteristic kernels include:

- Gaussian/RBF kernel: $k(x,y) = \exp\left(-\frac{\|x-y\|^2}{2\sigma^2}\right)$

- Laplacian kernel: $k(x,y) = \exp\left(-\gamma\|x-y\|_1\right)$

- Inverse multiquadratic kernels: $k(x,y) = (c^2 + \|x-y\|^2)^{-\beta}$ with $\beta > 0$

All these kernels can be expressed as expectations over random feature maps.

## B PROOF OF THEOREM 2.5

Let us begin by introducing the following probability flow ODE:

$$\dot{X}_t^* = b_t^*(X_t^*), \qquad X_{t=0}^* = z \sim \gamma, \tag{32}$$

where

$$b_t^*(x) = \langle \nabla\Phi(X_t^*), \eta_t^* \rangle \tag{33}$$

with $\eta_t^*$ solution to the linear system

$$\left\langle \mathbb{E}_{X_t^*}\left[\langle \nabla\Phi(X_t^*), \nabla\Phi(X_t^*)\rangle\right], \eta_t^* \right\rangle_{\mathcal{F}} = \mathbb{E}_{I_t}[\langle \nabla\Phi(I_t), \dot{I}_t\rangle] \tag{34}$$

Note that the expectation defining the operator at the right hand side in this equation is over $X_t^*$, the solution to (32): in contrast the expectation defining the operator $K_t$ appearing in (34) is over $I_t$.

Consider the MMD between the distributions of $I_t$ and the solution $X_t^*$ to (32):

$$\left\|\mathbb{E}_{I_t}[\Phi(I_t)] - \mathbb{E}_{X_t^*}[\Phi(X_t^*)]\right\|_{\mathcal{F}}^2 \tag{35}$$

By taking the derivative with respect to $t$ of this expression we obtain

$$\frac{d}{dt}\left\|\mathbb{E}_{I_t}[\Phi(I_t)] - \mathbb{E}_{X_t^*}[\Phi(X_t^*)]\right\|_{\mathcal{F}}^2$$

$$= \left\langle \left(\mathbb{E}_{I_t}[\Phi(I_t)] - \mathbb{E}_{X_t^*}[\Phi(X_t^*)]\right), \left(\mathbb{E}[\langle\nabla\Phi(I_t),\dot{I}_t\rangle] - \mathbb{E}_{X_t^*}[\langle\nabla\Phi(X_t^*),\dot{X}_t^*\rangle]\right)\right\rangle_{\mathcal{F}}$$

$$= \left\langle \left(\mathbb{E}_{I_t}[\Phi(I_t)] - \mathbb{E}_{X_t^*}[\Phi(X_t^*)]\right)\left(\mathbb{E}_{I_t}[\langle\nabla\Phi(I_t),\dot{I}_t\rangle] - \langle\mathbb{E}_{X_t^*}[\langle\nabla\Phi(X_t^*),\nabla\Phi(X_t^*)\rangle], \eta_t^*\rangle_{\mathcal{F}}\right)\right\rangle_{\mathcal{F}}$$

$$= 0$$

$$\tag{36}$$

where we used the chain rule to get the first equality, (32) to get the second, and (34) to get the last. By integrating this equation, we deduce

$$\forall t \in [0,1] \;:\; \left\| \mathbb{E}_{I_t}[\Phi(I_t)] - \mathbb{E}_{X_t^*}[\Phi(X_t^*)] \right\|_{\mathcal{F}}^2 = C \tag{37}$$

where $C$ is an integration constant. Since $X_{t=0}^* = z = I_{t=0}$, we know that the left hand side of (37) is zero at $t = 0$, i.e. $C = 0$ in (37). Since the kernel $k$ associated with $\Phi$ is characteristic by assumption, this implies that $X_t^* \overset{d}{=} I_t$ for all $t \in [0,1]$, which has two consequences:

First, the ODE (32) allows us to transport samples $X_t^*$ that have the same law as $I_t$, i.e. the velocity field $b_t^*(x)$ defined in (33) is an exact representation of the desired $b_t(x) = \mathbb{E}[\dot{I}_t | I_t = x]$.

Second, in (34) we can set $\mathbb{E}_{X_t^*}\left[\langle \nabla\Phi(X_t^*), \nabla\Phi(X_t^*)\rangle\right] = \mathbb{E}_{I_t}\left[\langle \nabla\Phi(I_t), \nabla\Phi(I_t)\rangle\right] = K_t$, i.e. (34) is the same equation as (9), meaning that $\eta_t^* = \eta_t$ and as a result the velocity field defined in (8) is the same as $b_t^*(x) = \mathbb{E}[\dot{I}_t | I_t = x]$.

To prove that (10) gives an exact representation of the score, we can follow similar steps starting from a probability flow ODE similar to (32) in which we express the velocity field in terms of the score using (6)

$$\dot{X}_t^* = \frac{\alpha_t s_t^*(X_t^*) + \dot{\beta}_t x X_t^*}{\beta_t(\alpha_t \dot{\beta}_t - \dot{\alpha}_t \beta_t)}, \qquad X_{t=0}^* = z \sim \gamma, \tag{38}$$

where

$$s_t^*(x) = -\alpha_t^{-1}\langle \nabla\Phi(X_t^*), \zeta_t^* \rangle \tag{39}$$

with $\zeta_t^*$ solution to the linear system

$$\left\langle \mathbb{E}_{X_t^*}\left[\langle \nabla\Phi(X_t^*), \nabla\Phi(X_t^*)\rangle\right], \zeta_t^* \right\rangle_{\mathcal{F}} = \mathbb{E}_{I_t}[\langle \nabla\Phi(I_t), z\rangle] \tag{40}$$

These calculation show again that the MMD distance between the solution to (38) is zero, implying that (i) (39) is an exact expression for the score and (ii) (40) is the same equation as (40), i.e. (10) is also an exact expression for the score. $\qquad\square$

Note that this proof shows that we can use (32) as alternative generative model. However (32) is more costly to integrate than the model with the velocity field and score specified in Theorem 2.5 since it requires the on-the-fly computation of $\mathbb{E}_{X_t^*}\left[\langle \nabla\Phi(X_t^*), \nabla\Phi(X_t^*)\rangle\right]$ to obtain $\eta_t^*$ from (34); in contrast the factor $\mathbb{E}_{I_t}\left[\langle \nabla\Phi(I_t), \nabla\Phi(I_t)\rangle\right]$ entering (9) can be pre-computed from the data.

## C    DETAILS OF THE NUMERICAL EXPERIMENTS

### C.1    MNIST UNET

A sinusoidal time embedding (scaled for $t \in [0,1]$) is passed through a small MLP and injected into every UNetBlock by projecting it to channel size and additively conditioning feature maps after the first conv. The U-Net has two encoder stages ($32 \to 64$ channels) with $3 \times 3$ convs, BatchNorm, and ReLU, each separated by $2 \times 2$ max pooling, followed by a bottleneck (128 channels). The decoder mirrors this with two transposed-conv upsampling steps, skip connections from the encoder, and UNetBlocks that reduce concatenated features back to 64 and then 32 channels. A final $1 \times 1$ conv produces the single-channel output. Time is never concatenated as an image channel; instead, its embedding conditions all scales of the network, enabling the model (VelocityFieldImage) to predict a time-dependent velocity field from a 1-channel input.

### C.2    CELEBA UNET

We followed the architecture and training setting of (Martin et al., 2025), yet we trained the weak models for only 5 epochs over the train set.

