# OpenReview forum: "Training-Free Generative Modeling by Kernelizing Pretrained Diffusion Models"
_ICLR.cc/2026/Conference — Submitted to ICLR 2026_

### Official Review · Reviewer_Z1LV · 2025-10-21

**Soundness:** 2
**Presentation:** 3
**Contribution:** 2
**Rating:** 4
**Confidence:** 4

**Summary:**

This paper proposes a method for ensembling pretrained diffusion models that is inspired by kernel learning. From a kernel learning perspective, the key insight is to solve the score-matching problem within the class of functions that are representable in the form $\hat{b}_t(x) = \langle \nabla \Phi(x), \eta_t \rangle$, where $\Phi$ is a feature map and the inner product is in the induced RKHS. The authors show that the ground truth score is representable in such a form if the RKHS has a characteristic kernel and that the weights $\eta_t$ can be learned by solving a system of linear equations. In practice, the authors replace the gradients $\nabla \Phi(x)$ with the velocity fields or score functions of a set of pretrained diffusion models, which reduces the learning problem to learning a set of global weights for each model and then sampling using linear combinations of the pretrained velocities/scores. The authors show that their method can be used to ensemble collections of weak diffusion models and briefly experiment with cross-domain enhancement.

**Strengths:**

- The paper is well-written, and I was able to follow the key ideas without much trouble. I appreciate the use of colored boxes to signpost important results.
- Both model ensembling and kernel learning for diffusion models are interesting problems, and the authors propose a sensible algorithm for learning a diffusion model's score or velocity in an RKHS
- Using this method to ensemble weak diffusion models improves on the sample quality that one would obtain from any of the individual models.

**Weaknesses:**

My primary critique of this paper is that while the theory is interesting, in practice, the method seems to boil down to learning a set of global weights (i.e. independent of $x$) for each pretrained model and sampling using a linear combination of the pretrained models' velocity fields. As the authors note, there is no guarantee that these velocity fields are in fact gradients in practice and that the resulting kernel is characteristic. This makes the theory seem somewhat superfluous: Can one use this theory to make meaningful predictions about the behavior of the method's practical implementation? Does it add much value relative to beginning with the (admittedly heuristic) ansatz that one should ensemble a set of diffusion models by taking a linear combination of their velocity fields and simply deriving the solution to the learning problem under that ansatz (i.e. lines 218-236)?

It's also unclear to me whether this in fact is a sensible strategy. For instance, wouldn't it make more sense to learn space- and time-dependent weights $\eta^i_t(x)$? It seems to me that at sampling time, one should weight each pretrained model according to some measure of their "confidence" in their prediction at $x$. For instance, if each model has learned a unimodal distribution and one wishes to sample from a multimodal distribution, then $\eta^i_t(x)$ should be large for models whose mode is close to x.

Building on the previous point, the authors might consider adding a simple baseline for their method: Simply fix uniform weights $\eta^i_t = \frac 1 P$ and otherwise sample as in Algorithm 1. I wonder if the weak models in Section 3.1 can be interpreted as noisy estimates of the score/velocity field near convergence, which can be "denoised" by simply averaging the estimates.

**Questions:**

- I would appreciate if the authors would comment on the novelty of their theoretical results for learning a diffusion model's score or velocity field in an RKHS -- particularly Theorem 2.5. I am not intimately familiar with the intersection of kernel learning and diffusion models, but the authors state in the related work that "RKHS reformulations of diffusion models have been considered e.g. in Maurais & Marzouk (2024); Yi et al. (2024): in contrast with these approaches that use standard kernels (Gaussian/RBF, Laplacian, etc.) we build the kernel using pre-trained score or drifts..." Do results similar to Theorem 2.5 appear in any of these prior works? If so, does this work's novel contribution boil down to using pretrained diffusion models as the source of the $\nabla \Phi$?

- Are there any notable theoretical obstacles to learning space- and time-dependent weights $\eta^i_t(x)$ in the proposed kernel learning framework?

- In Section 3.1 and Figure 3, the weak models are trained on Fashion-MNIST, EMNIST, MNIST, and Kuzushiji-MNIST, but the ensembled model's samples appear to be drawn exclusively from MNIST. Why is that? Is there any theoretical reason why we wouldn't expect the ensembled model to sample from any of the other data distributions? Is there any way to control which of the data distributions the ensembled model samples from?

---

> ### Author Response · Authors · 2025-11-24
>
> We thank the reviewer  for their insightful questions.
>
> **On whether the theory is superfluous:** The reviewer notes that in practice the method reduces to learning global weights for each pretrained model. While this is true operationally, the kernel formulation provides principled justification for which linear combination to use. The weights are derived from minimizing an objective (Equations 12-13) with theoretical guarantees under appropriate conditions---not chosen arbitrarily. Without this framework, one would need alternative justification for why linear combinations of velocity fields yield valid generative models.
>
> **On uniform weights baseline:** This is a useful suggestion that we will include. We expect learned weights to outperform uniform weights, particularly when weak models vary in quality or capture different aspects of the distribution, but the comparison will be informative.
>
> **On space/time-dependent weights:** This is an interesting direction but please note that extending to local weights $\eta_i(t,x)$ would require training these weights (using e.g. neural networks), substantially increasing computational cost. Our approach precisely avoids this and allow us to obtain $\eta_i(t)$ directly by solving a linear system once and for all.
>
> **On novelty of Theorem 2.5:** Prior works (Maurais & Marzouk, 2024; Yi et al., 2024) use standard kernels requiring explicit feature map construction and gradient computation. Our key observation is that pretrained models already provide these gradients directly (as scores or velocity fields), allowing us to construct expressive kernels without access to underlying feature representations. This specific construction enabling training-free model combination is, to our knowledge, novel.
>
> **On cross-domain results (Figure 3):** The ensemble samples from MNIST because the target distribution in Algorithm 1 is specified through MNIST data points ${a_1,...,a_N}$. The source-domain models (Fashion-MNIST, EMNIST, Kuzushiji-MNIST) provide complementary gradient information that improves generation quality for this MNIST target. To sample from a different distribution, one would change the target data. Controlling mixtures of distributions through the framework is an interesting extension we will investigate.

---

> > ### Comment · Reviewer_Z1LV · 2025-11-26
> >
> > Thanks for your helpful response to my review! I appreciate the clarification on the cross-domain results, which I misunderstood, and I eagerly await the results for the uniform weights baseline. I'm still not sure if I agree that the kernel formulation provides a sufficiently principled justification for the method. As you note, the weights coming from minimizing the objective in Eqs 12-13 come with theoretical guarantees if the kernel is characteristic. However, in practice, one cannot guarantee that the kernel arising from the pretrained velocity fields is characteristic, so the theoretical guarantees do not apply. I agree with the intuition that the span of pretrained velocity fields is a reasonable function space within which we might approximate the ground truth score, but I'm just not sure that the RKHS-based results provide much additional insight.
> >
> > *On space/time-dependent weights: This is an interesting direction but please note that extending to local weights $\eta_i(t,x)$ would require training these weights (using e.g. neural networks), substantially increasing computational cost. Our approach precisely avoids this and allow us to obtain $\eta_i(t)$ directly by solving a linear system once and for all.*
> >
> > This isn't especially obvious to me. Why would local weights necessarily require training? One heuristic that comes to mind is: Compute the probability of x at time t under the diffusion model parametrized by each pretrained velocity field (using e.g. the continuous change of variables formula from Song et al. 2021) and weight the velocity fields according to the relative probability they assign to x. This seems like a reasonable Bayesian model averaging scheme that doesn't require learned weights.
> >
> > I maintain my score for now.

---

### Official Review · Reviewer_WxFt · 2025-10-31

**Soundness:** 3
**Presentation:** 2
**Contribution:** 1
**Rating:** 2
**Confidence:** 3

**Summary:**

This paper proposes kernelization theories of diffusion models for training-free generative modeling. By leveraging Maximum Mean Discrepancy (MMD) and stochastic interpolants, the authors develop a kernelization framework that enables efficient sampling from pretrained diffusion models without additional training. The paper discusses two application scenarios: mixture of (weak) experts and cross-domain enhancement. Experimental results on standard datasets demonstrate the effectiveness of the proposed kernelization approach in generating high-quality samples.

**Strengths:**

- The background of kernelization and MMD is well explained, making the paper accessible.
- The kernelization framework is theoretically well-founded, providing novel perspectives on diffusion models.
- The discussed application scenarios of mixture of (weak) experts and cross-domain enhancement are sound and promising.

**Weaknesses:**

- The theoretical contributions, while solid, may lack depth in terms of novelty.
- The application scope of the kernelization framework is limited.
- The experimental validation is somewhat preliminary, requiring more extensive empirical evidence.

**Questions:**

1. **Theoretical novelty.** While the kernelization framework is well-founded, the kernalization theories, MMD definitions and stochastic interpolants have been extensively studied in prior works. The contribution of this paper seems to be an application or extension of these existing theories to diffusion models. Could the authors clarify the novel theoretical contributions of this work compared to prior literature?

2. **Application scope.** The proposed kernelization framework is applied to mixture of (weak) experts and cross-domain enhancement. However, these applications seem somewhat limited in scope. Could the authors discuss potential extensions or broader applications of the kernelization framework in diffusion models?

3. **Insufficient experimental validation.** While the authors discuss two potential applications, the experiments are performed on relatively simple datasets (e.g., MNIST). Could the authors provide more extensive experimental validation on larger and more complex datasets to better demonstrate the effectiveness of the proposed framework?

---

> ### Author Response · Authors · 2025-11-24
>
> We thank the reviewer  for their feedback.
>
> **On theoretical novelty:** While kernelization, MMD, and stochastic interpolants have been studied separately, our contribution lies in their specific combination: showing that pretrained velocity fields or scores can serve directly as kernel gradients, bypassing explicit feature map construction entirely. Prior RKHS formulations of diffusion models (Maurais & Marzouk, 2024; Yi et al., 2024) use standard kernels (Gaussian/RBF, Laplacian) requiring feature map identification and gradient computation. Our approach enables training-free combination of existing models—a capability not present in these prior works.
>
> **On application scope:** Beyond the two applications demonstrated (mixture of weak experts, cross-domain enhancement), the framework naturally extends to: conditional generation by incorporating conditional pretrained models, model adaptation with limited target data, and privacy-preserving settings where models cannot share parameters directly. We will expand this discussion in the revision.
>
> **On experimental validation:** We are conducting additional experiments on larger-scale datasets including ImageNet with standard quantitative metrics. We are confident these will demonstrate the broader applicability of our approach, and will present them in a revised version.

---

### Official Review · Reviewer_gAia · 2025-10-31

**Soundness:** 2
**Presentation:** 3
**Contribution:** 2
**Rating:** 4
**Confidence:** 4

**Summary:**

This paper provides a novel and interesting training-free generative modeling approach by kernelizing pre-trained diffusion models. The authors construct a kernel directly on the gradients of kernel features, leveraging pre-trained drift or score functions. The experiments demonstrate that the proposed approach can produce better generations by ensembling weak models. The method also shows successful cross-domain enhancement

**Strengths:**

* The paper proposes an interesting training-free generative modeling method by kernelizing pre-trained drift or score functions, which is novel. The derivations are overall correct.
* The paper provides empirical evidence showing that the proposed method improves generation quality on CelebA and MNIST, and it displays interesting cross-domain enhancement within MNIST-related settings.

**Weaknesses:**

* The main weakness lies in the simplicity and roughness of the evaluation. The experiments are conducted only on toy scenarios, which weakens confidence in extending the method to larger or more practical settings.
* Even for MNIST and CelebA, the paper provides insufficient information and analysis of the method.

Minor issue that does not directly affect my rating:

* The right-hand side of Equation 6 (representing velocity by score) is incorrect.

**Questions:**

* The expressiveness of the kernel is crucial for practical performance. Even on MNIST, the authors should design experiments to show how the kernel choice or the chosen weak models influence the final results (currently, only Figure 1 Right shows how the number of weak models influences performance).
* How do domain gaps influence cross-domain enhancement? How can pre-trained models on MNIST, SVHN, CIFAR, or ImageNet cooperate to produce better results on other domains such as CelebA?
* As a training-free approach that claims to save computational budgets, can the authors extend experiments to ImageNet with pre-trained models?
* In lines 254–255, we need to compute $b_t(X_t)$ with all $\sum^P_{i=1}b^i_t(X_t)\eta_t^i$. Does this mean we still need to forward all weak components $b_t^i$?
* We need a batch size of $N$ data points in the target domain; how does $N$ influence the results?

---

> ### Author Response · Authors · 2025-11-24
>
> We thank the reviewer for their constructive feedback and specific questions.
>
> **On Equation 6:** We thank the reviewer for catching this error. We will correct it in the revision. Note however that the claim that the score can be expressed in terms of  the drift and vice-versa remains valid.
>
> **On kernel choice and weak model selection:** This is an important question. The diversity and quality of weak models both matter: models initialized with different random seeds capture complementary aspects of the target distribution. We will include ablations systematically varying these factors in the revision.
>
> **On domain gaps in cross-domain enhancement:** The effectiveness of cross-domain enhancement depends on shared low-level structure (edges, textures) and compatible image statistics between domains. Models trained on semantically related datasets (e.g., different handwritten character sets) transfer more effectively than unrelated domains. We will include experiments characterizing how domain similarity affects enhancement quality.
>
> **On extending to ImageNet:** We are conducting experiments on ImageNet using publicly available pretrained models. We are confident these will demonstrate scalability, though they require time to complete properly.
>
> **On forwarding all weak models at inference (lines 254-255):** Yes, at inference we evaluate all $P$ models at the current state. However, $P$ can remain small (10-25 in our experiments) while achieving significant improvements. The linear solve itself is negligible compared to model evaluation and is performed only once.
>
> **On batch size N:**
> The batch size $N$ affects the accuracy of the empirical expectations in the linear system (which, we stress, need to be solved only once). Larger $N$ reduces variance but increases memory requirements. We will include ablations on this parameter in the revision.

---

### Official Review · Reviewer_33k8 · 2025-11-01

**Soundness:** 3
**Presentation:** 2
**Contribution:** 2
**Rating:** 2
**Confidence:** 3

**Summary:**

This work proposes a method to construct a generative diffusion model in the (allegedly) *training-free* way, building upon existing pretrained, possibly weak, diffusion models. Theoretically, based on the stochastic interpolant model, the authors reduce the generative modeling task into finding an appropriate drift $b_t$ to be used in a certain SDE, and then suggest building this $b_t$ from predefined feature maps (or more precisely, their derivatives) using a kernel regression type method. Relying on the freedom to select which features are to be used, the paper chooses them to be the log-density functions (whose derivatives are the score functions) learned by pretrained diffusion models. Experiments are demonstrated, as evidence to support the validity of the claim.

**Strengths:**

The idea of leveraging Hilbert space kernel regression into finding the score function is interesting and noteworthy. The theoretical developments are sound, giving us concrete training recipes. The experiments, while preliminary, offer encouraging visual evidence supporting the approach.

**Weaknesses:**

Above all, the paper would benefit from articulating a well-defined objective, ideally with a clear mathematical formulation, in the introduction and maintaining a consistent focus on it throughout. For me, it was not entirely clear whether the ultimate aim of this paper is to propose a new fine-tuning strategy, a novel method for obtaining score functions, an approach to distributed training, or some combination of these.

After reading the paper in full, my understanding is that the main goal of this paper aligns with what I described in the **Summary** section. It is a promising direction, and the core idea is a very interesting one. However, the theoretical contributions are incremental, being no more than a special case of well-known facts in kernel regression theory. From the practitioner's point of view, it seems several challenges still need to be addressed. Please see the **Questions** section below. I would be more than happy to have a constructive discussion with the authors, and I am willing to increase my score accordingly if my concerns are adequately addressed.

**Questions:**

1.  Theorem 2.5, which is central to this work, relies on a relatively strong assumption that the kernel is *characteristic*. This is not that of a big problem when we have full freedom to choose which kernel, or equivalently the features, to use. However, once we get into the "weak-to-strong" tasks (building the drift function $b_t$ using weak pretrained models), we no longer can guarantee that the kernel is characteristic, and being able to recover $b_t$ only makes sense if $b_t$ is in the span, or at least very close to the span, of $\nabla \Phi_i$s. The only scenarios I can think of this happening is either (1) we use a massive amount of pretrained models so that the span gets sufficiently large, (2) when the "weak" models are actually already expressive enough (although this seems somewhat contrary to the set-up under consideration), or (3) the dataset lies in a very low intrinsic dimension, enough to guarantee that the true drift varies primarily in a low-dimensional subspace, so that a small ensemble of weak model suffices. Otherwise, the learned drift will be the projection of $b_t$ onto the span of $\nabla \Phi_i$, which can be very different from what we want to learn. This seems to be a critical gap between theoretical explanation and the actual proposed method, and I would like to hear how authors think about this.

2. I can see where the naming *"training-free"* comes from; the "training" part, in practice, reduces to solving a system of linear equations. However, it seems like the training cost is just transferred to the inference part; now during inference, we have to get inference results from all of the weak models, in order to take the linear combination of them. Can we say for sure that this approach has a clear advantage over paying the cost once in the training phase?

3. I am quite skeptical about the validity of the experimental results for a few reasons: (1) there are no quantitative results regarding the sample quality and the report only relies on the visual aspects, (2) only a handful of samples are shown for each experiments, not fully convincing that the model is performing well, and (3) the experiments are only on "relatively easy" datasets such as MNIST and Celeb-A. Testing on easier datasets is not that of a big problem, as they can serve as a proof-of-concept for a theoretical result, but the first two points are much more critical and should be discussed carefully when doing so, which the paper currently lacks. Could the authors explain why these decisions were made when presenting the experimental results, and are there any plans to provide further elaboration or quantitative evaluation?

---

> ### Author Response · Authors · 2025-11-24
>
> We thank the reviewer  for their careful reading and thoughtful questions.
>
> **On the characteristic kernel assumption:**
> The reviewer outlines three scenarios where our theory applies: (1) a massive number of pretrained models, (2) weak models that are already expressive enough, or (3) data lying in low intrinsic dimension. While all three are potential applications of our approach, scenario (2) is the one we investigated in our paper. Pretrained diffusion models are designed to be individually expressive enough to represent the target distribution. A collection of such models therefore spans a function space that is sufficiently rich for practical purposes. Moreover, our framework optimizes over their best features, yielding better accuracy than any individual model. This is precisely what we observe empirically: ensemble performance consistently improves with the number of models and exceeds that of any single component.
>
> **On training cost transferred to inference:** This is a valid concern. However, the linear systems need only be solved once and can be reused across all sample generations. The per-sample inference cost involves evaluating $P$ pretrained models at each timestep, but $P$ can remain small (10-25 in our experiments) while achieving significant improvements. We will include a detailed computational analysis comparing training costs of strong models versus inference costs of our ensemble approach in the revision.
>
> **On experimental validation:** We agree that our current experiments serve primarily as proof-of-concept. We are conducting additional experiments with standard quantitative metrics (FID, improved likelihood estimation) on larger-scale datasets. We are confident these will further demonstrate the utility of our approach. However, these numerical studies require time to complete properly, and we will present them in a revised version.

---

### Author Response · Authors · 2025-11-24

We thank all reviewers for their careful reading of our manuscript and their constructive feedback.

We would like to address some of the main concerns raised:

**On the characteristic kernel assumption (Reviewers 33k8, Z1LV):** We respectfully disagree that this represents a critical gap. Pretrained diffusion models are designed to be individually expressive enough to represent the target distribution. A collection of such models therefore spans a function space that is, for practical purposes, sufficiently rich. Indeed, the kernel constructed from their gradients should be "effectively characteristic" for distributions of interest---and our framework optimizes over their best features, yielding better accuracy than any individual model. This is precisely what we observe empirically: ensemble performance consistently improves with the number of models and exceeds that of any single component. We will clarify this reasoning in the revised manuscript.

**On experimental validation (All reviewers):** We agree that our current experiments serve primarily as proof-of-concept. We are currently conducting additional experiments with standard quantitative metrics (FID, improved likelihood estimation) on larger-scale datasets. We are confident these will further demonstrate the utility of our approach. However, these numerical studies require time to complete properly, and we will present them in a revised version.

**On computational trade-offs (Reviewer 33k8):** The linear systems need only be solved once and can be reused across all sample generations. The inference cost thus depends on the number P of pretrained models, which our experiments show can remain small (10-25 models) while achieving significant improvements. We will include a more detailed computational analysis in the revision.

**On the uniform-weights baseline (Reviewer Z1LV):** This is a useful suggestion that we will include. We note, however, that uniform weighting lacks theoretical grounding and does not account for the varying quality and complementarity of different models—precisely what our kernel formulation addresses.

**On cross-domain results (Reviewer Z1LV):** The ensemble samples from MNIST because the target distribution in Algorithm 1 is specified through MNIST data points. The source-domain models provide complementary gradient information that improves generation quality for this target. We will clarify this in the revision.

We remain confident in the potential of our approach and will provide a substantially expanded revised version addressing the points above.

---

### Meta-Review · Area_Chair_Lm7k · 2026-01-09

**Summary:**

Reviewers 33k8 and Z1LV point limits to the main result of the paper; 33k8 points a limit in the evaluation of the experiments with respect to the metric used (same for gAia but with a specific experiment suggested); WxFt points out too simple experiments;

**Reviewer Concerns:**

There was no additional experiment provided and the theoretical justifications were a bit shallow

**Reviewer Scores:**

all reviews have a similar negative polarity and aligned on several problems the paper has, it is unlikely things would have changed with deeper discussion participation.

---

### Decision · Program_Chairs · 2026-01-26

Reject